# Predictors of Complicated Appendicitis with Evolution to Appendicular Peritonitis in Pediatric Patients

**DOI:** 10.3390/medicina59010021

**Published:** 2022-12-22

**Authors:** Laura Bălănescu, Alexandru Emil Băetu, Ancuța Mihaela Cardoneanu, Andreea Alecsandra Moga, Radu Ninel Bălănescu

**Affiliations:** 1Pediatric Surgery Department, “Grigore Alexandrescu” Clinical Emergency Hospital for Children, 011743 Bucharest, Romania; 2Pediatric Surgery Department, “Carol Davila” University of Medicine and Pharmacy, 020022 Bucharest, Romania; 3Department of Anaesthesia and Intensive Care, “Grigore Alexandrescu” Clinical Emergency Hospital for Children, 011743 Bucharest, Romania; 4Department of Anaesthesia and Intensive Care, “Carol Davila” University of Medicine and Pharmacy, 020022 Bucharest, Romania

**Keywords:** appendicular peritonitis, novel biomarkers, neutrophil-to-lymphocyte ratio, platelet-to-lymphocyte ratio

## Abstract

*Background and Objecitves:* Appendicitis is one of the most frequent surgical emergencies in pediatric surgery. Complicated appendicitis can evolve with appendicular peritonitis characterized by the diffusion of the pathological process to the peritoneal cavity, thus producing generalized or localized inflammation of the peritoneum. The capacity to anticipate the possibility of perforation in acute appendicitis can direct prompt management and lower morbidity. There is no specific symptom that could be used to anticipate complicated appendicitis, and diagnostic clues include a longer period of symptoms, diffuse peritoneal signs, high fever, elevated leukocytosis and CRP, hyponatremia, and high ESR. Imagistic methods, particularly US and CT, are useful but not sufficient. There are no traditional inflammation biomarkers able to predict the evolution of uncomplicated to complicated appendicitis alone, but the predictive capacity of novel biomarkers is being investigated. *Materials and Methods:* The present study represents a retrospective evaluation of children hospitalized between January 2021 and July 2022 in the Grigore Alexandrescu Clinical Emergency Hospital for Children with a diagnosis of acute appendicitis settled based on clinical characteristics, traditional and novel biomarkers, and ultrasonographic features. The children were subsequently grouped into two groups based on the existence of appendicular peritonitis on intraoperative inspection of the abdominal cavity. The aim of this report is to establish the predictors that may aid physicians in timely identifying pediatric patients diagnosed with acute appendicitis at risk for developing complicated appendicitis with evolution to appendicular peritonitis. *Results:* The neutrophil-to-lymphocyte ratio (NLR) and platelet-to-lymphocyte radio (PLR) are representative severity markers in infections. This report analyzes the benefit of these markers for distinguishing uncomplicated appendicitis from complicated appendicitis in pediatric patients. *Conclusions:* Our study suggests that a value of neutrophil-to-lymphocyte ratio greater than 8.39 is a reliable parameter to predict the evolution to appendicular peritonitis.

## 1. Introduction

Acute appendicitis is one of the most frequent surgical emergencies in both the pediatric and adult populations [1]. Despite advances in the field of laboratory and radiological modalities, the diagnosis is mainly based on classical presentation and physical examination [2]. The diagnosis of acute appendicitis in pediatric patients is challenging, as many of them do not have typical clinical manifestations, especially in younger children, especially infants who present with nonspecific signs such as irritability, anorexia, and lethargy [3]. The delay in diagnosing acute appendicitis can result in an increased risk of perforation and further complications such as bacterial peritonitis, bowel obstruction, and abdominal fluid collection, leading to a longer hospital stay and a higher mortality rate [3]. Therefore, it is very important for pediatric surgeons to enhance the initial diagnosis of perforated appendicitis accurately. Complicated appendicitis is defined by the presence of perforations of the appendix anyplace from the tip to base with abscess development or progression to acute peritonitis, which accounts for approximately 30% of cases [4,5]. Identifying an instrument or marker that can anticipate the diagnosis of acute appendicitis and may differentiate uncomplicated and complicated appendicitis with acceptable sensitivity and specificity is, up to this day, an object of concern among researchers.

The predictive ability of risk scores, serum biomarkers, and imaging modalities such as transabdominal ultrasonography (US), abdominal computed tomography (CT), or magnetic resonance imaging (MRI) techniques might aid the clinician in choosing the best medical and surgical management strategy for these patients. Numerous scoring systems have been proposed in order to support the clinician in diagnosing and categorizing acute appendicitis, such as the Alvarado score, pediatric appendicitis score (PAS), the appendicitis inflammatory response (AIR), and Shera score. These scoring systems lack the sensitivity and specificity to anticipate the severity of acute appendicitis; thus, their use is not recommended by the WSES [6].

A complete blood count (CBC) is the principal laboratory method to diagnose appendicitis. Inflammation biomarkers commonly assessed include total and differential leukocyte count, C- reactive protein (CRP), or procalcitonin, but all of these markers lack the sensitivity and specificity to be used alone in making the diagnosis of appendicitis. The accuracy of white blood cells (WBC) and neutrophils for diagnosis of appendicitis differ among articles, as normal leukocyte counts do not exclude appendicitis, and CRP is a sensitive test, but non-specific for acute appendicitis [2,7]. However, CRP and procalcitonin, as well as novel derived biomarkers such as neutrophil-to-lymphocyte ratio (NLR), platelet-to-lymphocyte ratio (PLR), and monocyte-to-lymphocyte ratio (MLR) were successfully used as severity markers in predicting complicated appendicitis in children [8,9,10,11]. These are easy, inexpensive markers of inflammation that are obtained easily. NLR is a new inflammatory biomarker and its value rests on its readily available nature, is calculated from CBC, is inexpensive, and is easily applicable [8].

NLR offers data about two different immune and inflammatory pathways, which makes it a possible indicator to predict appendicitis and its severity [9]. The neutrophil count indicates active and continuing inflammatory changes, while the lymphocyte count highlights the regulatory pathway [12,13,14].

The accessible literature indicates that an NLR of 4.7 is a cut-off value for uncomplicated appendicitis and 8.8 for complicated appendicitis, with a sensitivity ranging from 62–92% and a specificity of 56–89% [6]. 

NLR was not integrated into the updated 2020 WSES guidelines, but because of its simplicity and availability in various clinical settings, it can be successfully used as a supplementary marker for diagnosing pediatric appendicitis and its evolutive forms [11].

In children with suspected acute appendicitis, the intent is to achieve essential surgical management without delay and to avoid unneeded appendectomy. For this aim, imaging methods are often used in doubtful patients [15,16,17].

This study investigates the clinical characteristics, traditional and novel biomarkers, and ultrasonographic features between cases of uncomplicated and complicated appendicitis with evolution to appendicular peritonitis. The aim of this study is to determine the predictors that may aid physicians in timely identifying pediatric patients diagnosed with acute appendicitis at risk for developing complicated appendicitis with evolution to appendicular peritonitis.

## 2. Materials and Methods

The present study proposes a retrospective study of children hospitalized between January 2021 and July 2022 in the Grigore Alexandrescu Clinical Emergency Hospital for Children with the diagnosis of acute appendicitis established on clinical examination, laboratory analyses, and imaging evaluations. The inclusion criteria were pediatric patients under the age of 18 years old who came to the emergency room with signs and symptoms of acute appendicitis and underwent appendectomy. Patients with the diagnosis of uncomplicated appendicitis and treated conservatively, patients with chronic inflammatory diseases, hematologic diseases, or autoimmune conditions, as well as patients with neoplasms of the appendix, and children with incomplete medical data, were excluded from our study. All data were obtained retrospectively from hospital database records. The clinical diagnosis was established by patient history, physical examination, traditional laboratory tests, and imaging studies. The pathological diagnosis was determined by intraoperative findings and confirmed by histopathological examination of the resected appendix; thus, 599 patients under the age of 18 are included in the study. The children were subsequently grouped into two groups conditioned by the diagnosis of complicated or uncomplicated appendicitis. The diagnosis of uncomplicated appendicitis was established as congestive or catarrhal, phlegmonous or suppurative, gangrenous, or necrotic. Complicated appendicitis was defined by the presence of perforations of the appendix anyplace from the tip to base, abscess development, or the presence of peritonitis, aspects that were established on intraoperative inspection of the abdominal cavity and confirmed by histopathological examination of the specimen.

All patients went through an accurate anamnestic survey, physical examination, establishment of body temperature, PAS score assessment, and standard laboratory tests. Abdominal ultrasound was also performed, and the diameter of the appendix was determined, as well as the existence of fecalith, periappendiceal effusion, periappendiceal abscess, and free intra-abdominal fluid. All patients had biological samples collected at the time of admission for obtaining the usual laboratory analyses: hemogram and inflammatory samples (C-reactive protein, erythrocyte sedimentation rate, and fibrinogen). Novel inflammation biomarkers were subsequently calculated from the hemogram: NLR, PLR, MLR, and hemoglobin-to-platelets ratio (HPR). On admission, all patients were examined by an experienced pediatric surgeon, and based on the clinical judgment, the children were classified as those to be operated on and those to be treated conservatively.

Statistical analysis was performed with the GraphPad 9 program. Data normality was tested using the D’agostino Pearson test.

The approval of the ethics committee of the Children’s Emergency Clinical Hospital was obtained both in terms of data collection and analysis and in terms of publishing the results (no. 27953/28.09.2022).

## 3. Results

The study included 599 patients under the age of 18, of whom 299 are girls and 370 are boys. After the intraoperative inspection of the abdominal cavity, 164 patients were diagnosed with complicated appendicitis and appendicular peritonitis (group A), and 435 with acute uncomplicated appendicitis (group B).

The mean age of patients with appendicular peritonitis (135.5 ± 48.36 months) was not statistically different from the mean age of patients with appendicitis (129.3 ± 48.36 months) *p* = 0.12. In contrast, the median time to onset of typical symptoms was longer in patients with peritonitis compared to patients with appendicitis (Mann–Whitney 48 h vs. 24 h *p* < 0.0001).

Regarding the classic panel of paraclinical analyses, there are statistically significant differences between the average number of leukocytes, neutrophils, lymphocytes, and platelets of patients with appendicular peritonitis compared to patients with uncomplicated acute appendicitis (Table 1).

Of the new predictive biomarkers tested, only the neutrophil-to-lymphocyte ratio and the platelet-to-lymphocyte ratio differed statistically between the two study groups. Thus, patients with peritonitis have a higher NLR compared to those with acute appendicitis and a lower PLR compared to them (Table 2).

In order to determine cut-off points with their sensitivity and specificity, we made two ROC curves using as an independent variable, the diagnosis of appendicitis peritonitis and as dependent variables, NLR, PLR, CRP, fibrinogen, and ESR.

ROC analysis of the ratio of neutrophils to lymphocytes between the two groups has an area under the curve of 0.77 95%CI 0.73 to 0.80 (*p* < 0.0001). The cut-off point with NLR > 8.39 has a sensitivity of 73.17% and a specificity of 70.11% (Figure 1, Table 3).

Regarding the ROC analysis of the ratio between platelets and lymphocytes, we obtained an area under the curve = 0.68 (AUC), with 95% CI of 0.64 to 0.74, *p* < 0.0001. The cut-off point with the best sensitivity (63.41%) and specificity (64.14%) was 201.4 (Figure 2, Table 3).

By comparison with the ROC curves of the other inflammatory samples (CRP-AUC = 0.72, fibrinogen-AUC = 0.70, ESR-AUC = 0.66), NLR remains a good predictor of appendicular peritonitis.

## 4. Discussion

Acute appendicitis is a frequent source of acute abdomen in children. Appendicitis contains a spectrum of manifestations, from simple inflammation of the appendix to perforated appendicitis with local or extensive contamination [14].

Children do not usually present the classic manifestations of appendicitis, and correct diagnosis is difficult. Traditionally, signs and symptoms such as right lower quadrant tenderness, rebound or percussion tenderness, migrated abdominal pain to the right lower quadrant, duration of abdominal pain, vomiting, anorexia, and increased body temperature have been used as criteria for different risk scores used as screening or prognostic tools [15].

Acute appendicitis is the most frequently misdiagnosed cause of acute abdomen pain in children due to the delay or misdiagnosis being attributed to non-specific symptoms, overlapping of symptoms with other pathologies, and the difficult abdominal examination, especially in younger patients [8]. The lag in diagnosis carries with it a high risk of gangrene, perforation, intra-abdominal abscess formation, peritonitis, and sepsis, thus increasing morbidity rates [2,16]. The surgeon has a hard choice between waiting to perform surgery until a complete diagnosis and operating soon after diagnosis to elude complications [17,18]. The preoperative difference between complicated and uncomplicated appendicitis may be difficult [19,20]. Clinical examination identifying signs of peritoneal irritation and imaging studies showing signs of complicated conditions such as intra-abdominal abscesses, pneumoperitoneum, or free intra-abdominal liquid is useful, but the definitive diagnosis of perforated appendicitis still requires surgery and histological diagnosis [3,21,22]. Consistently throughout the literature, there are several factors indicating a high risk of perforated appendicitis, such as younger age at presentation, a longer duration of symptoms, and some non-specific manifestations such as anorexia and emesis [18,23]. Diffuse pain, generalized abdomen tenderness, fever, and peritoneal signs are most frequently present in perforated appendicitis compared to simple appendicitis [14]. The distinction between uncomplicated and complicated appendicitis guide the surgeon in establishing the severity and the emergency of the case and can also be useful in advising the parents regarding the postoperative course, morbidity, and length of hospital stay [22,24]. Pediatric perforated appendicitis rates are reported at approximately 30%, ranging from 20% to 74%, but may be higher in young patients [17,18]. The existence of perforation has a major role in children’s morbidity, with perforated appendicitis being associated with an elevated risk of upcoming postsurgical complications such as intra-abdominal abscesses, wound infections, postoperative ileus pelvic fluid accumulation, and elevated rates of readmissions and longer length of hospital stay [18,25].

The latest studies reported that pediatric patients with uncomplicated appendicitis can be amenable to nonoperative treatment with antibiotics alone. Hence, it is essential to be able to anticipate which patients may evolve into complicated appendicitis as these children have to undergo surgical treatment and not be proposed for nonoperative strategy [22]. However, since appendicular perforation has an important influence on morbidity, mortality, and postoperative outcome, it is essential to identify it soon and remove it surgically. While some authors approve of a certain rate of negative appendectomies to overcome morbidity and mortality due to perforation, others consider it unacceptable because of the morbidity and mortality associated with the surgery itself [23]. Negative appendectomy rates were reported to be under 10% in the last decade due to improved diagnostic methods [24].

Although complete clinical judgment, laboratory findings, and imaging modalities are usually necessary to make the diagnosis, all have limitations. In this regard, a search for some other diagnostic parameters was necessary [8].

Complete blood count (CBC) is an essential part of diagnosis in children with suspicion of appendicitis, leukocytosis, C-reactive protein (CRP), erythrocyte sedimentation rate (ESR), and hyponatremia can help the diagnosis [14]. Even if leukocyte count is usually elevated in patients with acute appendicitis, it is not a specific marker for acute appendicitis, and it can be increased in other diseases related to other inflammatory conditions considered in the differential diagnosis [16]. Gosain et al. reported that CRP and leukocytosis are reliable predictors of perforated appendicitis, but only leukocytosis higher than 19.400 cells per microliter was a multivariate predictor for perforation [25]. Serum CRP level is the most used marker of the acute phase protein. It was proved that in children with symptoms that had lasted less than 24 h, WBC count had a high sensitivity, while in children in whom they had lasted more than 24 h, CRP had a higher sensitivity [26]. Anyway, CRP and WBC values can be normal in 8% of children with proven appendicitis [26]. Hyponatremia has been reported as a strong independent predictor of complicated appendicitis in pediatric patients [14].

Clinical examination and laboratory markers have been conjugated in the form of appendicitis clinical risk evaluation scores (Alvarado score, appendicitis inflammatory response score- AIR, and RIPASA score) to help the clinician by stratifying the risk and influence the following investigations and interventions [27]. However, the scores are commonly lower in pediatric patients because they are not cooperative or they are too young to be able to explain their complaints. These scoring systems are also not able to distinguish simple appendicitis from complicated ones. The results of appendicitis scoring systems are still controversial among different reports [28].

Imaging is a useful adjunct to the diagnosis of appendicitis, but it is not mandatory, and the diagnostic accuracy is not 100%. Abdominal ultrasonography is favored due to its low-risk profile, and it can be helpful in diagnosis, but it is operator-dependent, and the presence of artifacts such as bowel gas or obesity is a challenging hurdle for diagnosis [2]. By means of abdominal ultrasonography and computed tomography, they look for possible positive signs such as increased appendix diameter, appendix wall thickening, periappendicular fatty tissue heterogeneity, or appendicoliths [12]. An appendix with a diameter higher than 6 mm has been described as significant for acute appendicitis in many articles. In our study, we found that appendix diameter was significantly higher in patients with complicated appendicitis.

Identifying minimally invasive diagnostic tools has always been of great importance, especially in pediatric medicine, and complete blood count-derived markers such as NLR, MLR, and PLR were demonstrated to be useful diagnostic and prognostic tools in assessing the systemic impact of the immune and inflammatory response in a wide range of conditions such as sepsis, trauma, malignancy, cardiovascular disease, or aging [29,30].

The pathophysiological explanation resides in the alteration of the signaling pathways and subsequent changes in microenvironment conditions in response to acute and chronic stress factors [31,32,33].

For any of the aforementioned ratios to increase, there must be either an increase of the numerator and/or a decrease of the denominator. The decrease in lymphocyte count is usually multifactorial and comprises alterations in cellular production, distribution, and turnover. Qualitative and quantitative abnormalities of the circulating lymphocytes and natural killer cells have been associated with immune dysfunction in patients with sepsis. Elevated levels of circulating neutrophils are affiliated with decreased activity of other immune cells, such as T-lymphocytes [34]. The ratio between neutrophils and lymphocytes boosts more quickly after the physiological stress than other laboratory markers, such as leukocytes. Even with normal values of leukocytes, this ratio has been reported to be a predictor of inflammatory processes [18]. There are different cytokine profiles associated with the upregulation of monocytes and neutrophils in complicated appendicitis and the upregulation of basophils and eosinophils in non-complicated appendicitis [35]. Platelets are key factors in the thrombo-inflammatory response and interact with leukocytes and monocytes, promoting cell signaling, migration, and even cell phenotypic changes [36]. Large and active platelets are more susceptible to being sequestered and consumed at sites of inflammation, conducting to a lower platelet count in sepsis and other severe types of inflammatory processes such as perforated appendicitis [17].

Recently, the interest in conservative management of uncomplicated acute appendicitis has grown, and NLR can play a role in children with radiologically confirmed uncomplicated appendicitis who are managed conservatively in terms of monitoring the response to nonoperative management, anticipating the risk of complications and recognizing the failure of nonoperative treatment [9]. Studies have shown that as the severity of appendiceal inflammation increases, lymphocyte counts decrease in addition to neutrophilia; consequently, NLR increases as appendicitis progresses to appendiceal gangrene and subsequent perforation [16].

Goodman et al. described neutrophil-lymphocyte ratio (NLR) as a diagnostic instrument for the first time, and when this ratio was higher than 3.5, they showed that it was significant in the diagnosis of acute appendicitis [37]. In the following years, other authors demonstrated that NLR is a marker of inflammation and found it to play a preoperative diagnostic role in uncomplicated and complicated appendicitis [16]. Markar et al. have demonstrated that NLR has higher diagnostic sensitivity than WBC or CRP alone in acute appendicitis and is an independent predictor of confirmed appendicitis histology [38]. Shimizu et al. reported that a greater NLR is closely associated with complicated appendicitis [39]. In a retrospective study by Kahramanca, they reported that an NLR cutoff value of 5.74 was found to be critical for complicated appendicitis [40]. Ishizuka demonstrated that NLR above 8 was significant for gangrenous appendicitis [41]. Khan et al. confirmed that an NLR > 6.36 or CRP > 28 was statistically associated with complicated acute appendicitis and NLR had a better area under the ROC curve compared to CRP for predicting appendicitis [42]. The diagnostic certainly of the NLR for distinguishing patients with complicated appendicitis has uncertain results in different studies [7]. The capacity of NLR to distinguish between simple and complicated appendicitis in pediatric patients, as measured by the AUC, ranges from 0.66 to 0.84, with cut-off points between 4.8 and 10.4 and a sensitivity of 67–85% [18]. Mori et al. demonstrated that an increased NLR is a good predictor of postoperative infections in children with acute appendicitis [43].

A meta-analysis including more than 8.000 adult patients identified the NLR cut-off value of 4.7 for the diagnosis of appendicitis (sensitivity of 88.89% and specificity of 90.91% with AUC of 0.96) and a cut-off value of 8.8 for complicated appendicitis (sensitivity of 76.92% and specificity 100% with AUC of 0.91) [9]. In our study, the cut-off point with NLR > 8.39 has a sensitivity of 73.17% and a specificity of 70.11%.

PLR is a merger of the PLT and lymphocyte counts. The analysis of PLR is focused on cancers and inflammation. Nazik et al. reported that NLR, PLR, and ESR values can be useful in the diagnosis of appendicitis [44]. Pehlivanli and Aydin reported that a PLR > 140.45 has a sensitivity of 71.4% and a specificity of 88.9% to discriminate between appendicitis and a normal appendix, whereas a PLR > 163.27 has a sensitivity of 64.3% and a specificity of 67.5% to differentiate complicated appendicitis from uncomplicated appendicitis [44,45]. However, our data suggest a higher cut-off value of 201.4, with a PLR sensitivity of 63.41% and a specificity of 64.14%. A retrospective single-center study assessed the use of NLR and PLR in predicting the severity of acute appendicitis in children, and their statistical analysis demonstrated both factors as independent predictors of complicated appendicitis. Using NLR and PLR AUC and cut-off was 0.776, 8.86 with a 95% CI of 0.730–0.822 and 0.694, 193.67 with a 95% CI of 0.634–0.755, respectively [6].

Statistical analysis of the data from our pediatric cohort demonstrated that of the new predictive biomarkers tested, only the NLR and the PLR differed statistically between the two study groups. Our data corresponds with the values in the literature. Thus, patients with appendicular peritonitis have a higher NLR compared to those with acute uncomplicated appendicitis and a lower PLR compared to them.

## 5. Conclusions

NLR and PLR ratios are useful tools that can anticipate both the diagnosis and severity of appendicitis in pediatric patients and, when interpreted together, could have acceptable sensitivity and specificity. Our study suggests that a value of neutrophil-to-lymphocyte ratio greater than 8.39 is a reliable parameter to predict the evolution to appendicular peritonitis. This value is consistent with those reported in previous studies. However, the strongly statistically significant values of the other inflammatory samples for the ROC curves give them a good predictive ability but with lower sensitivity and specificity than the ratio of neutrophils to lymphocytes. NLR is a simple, not expensive, quickly available, and affordable biomarker that can be used not only for the diagnosis of acute appendicitis but also for the prediction of the evolution to complicated appendicitis, particularly when there is atypical presentation, inadequate clinical findings, and no other laboratory variables are available in the emergency department. NLR can also be used to monitor children with appendicitis who are being managed nonoperatively.

## 6. Study Limitations

The limitations of our study include those that are inherent to its retrospective design, e.g., that this is a single-institution study that makes our findings not generalizable. A prospective observational study, including other centers, is required to assess the utility of our findings. However, we believe that our study provides comprehensive data on the diagnostic accuracy of simple laboratory parameters in the prediction of the evolution of acute appendicitis, and this study contributes to the literature with useful reference data. Although we recognize the limitations of our study, we believe that our data offer further validation of previously published data evaluating the utility of NLR and PLR in predicting appendicitis severity.

## Figures and Tables

**Figure 1 medicina-59-00021-f001:**
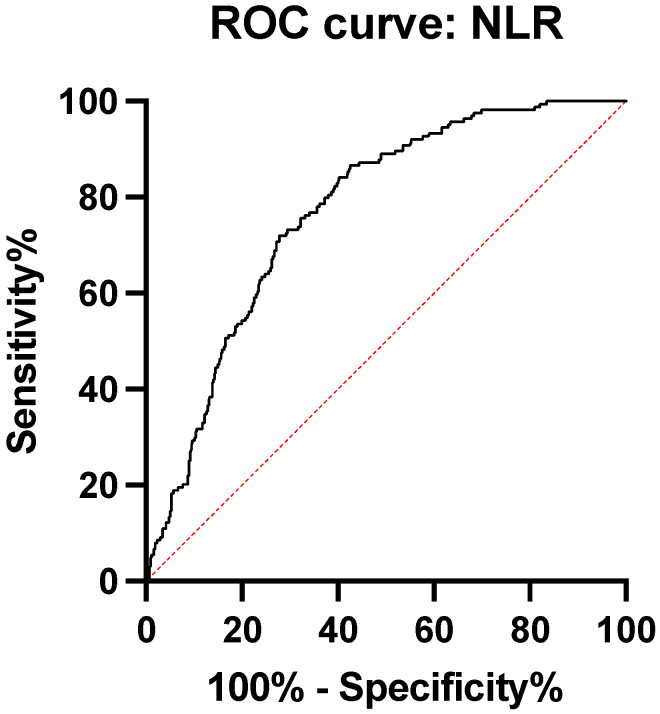
ROC curve neutrophil to lymphocyte ratio.

**Figure 2 medicina-59-00021-f002:**
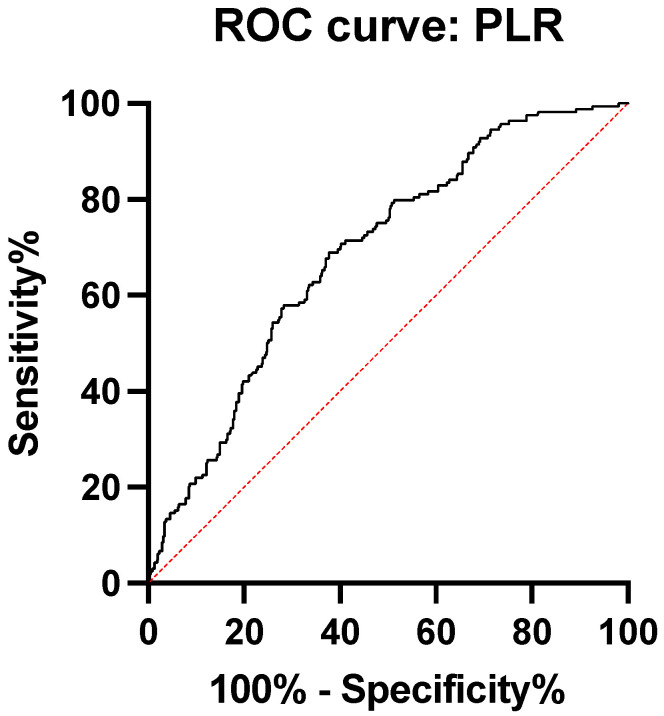
ROC curve platelets to lymphocyte ratio.

**Table 1 medicina-59-00021-t001:** Comparative analysis of paraclinical data obtained from hemogram and inflammatory samples.

	Mean Group A	Mean Group B	Difference between Means ± SEM	95% CI of the Difference	*p* Value
WBC	15.89	14.26	−1.6 ± 0.47	−2.56 to −0.7081	0.0006
Neutrophil	14.5	10.1	−4.39 ± 0.41	−5.2 to −3.5	<0.0001
Lymphocytes	1.65	2.38	0.72 ± 0.29	0.15 to 1.3	0.01
Platelets	327.1	310.4	−16.75 ± 7.43	−31.35 to −2.1	0.02
CRP	9.81	3.96	−5.85 ± 0.76	−7.36 to −4.34	<0.0001
Fibrinogen	464.6	352.2	−112.4 ± 12.34	−136.7 to −88.2	<0.0001
ESR	30.04	17.24	−12.81 ± 1.8	−16.35 to −9.26	<0.0001

**Table 2 medicina-59-00021-t002:** Comparative analysis of the novel biomarkers.

	Mean Group A	Mean Group B	Difference between Means ± SEM	95% CI of the Difference	*p* Value
NLR	13.02	7.13	−5.88 ± 0.59	−7.053 to −4.71	<0.0001
PLR	199.8	287.8	87.97 ± 13.54	61.38 to 114.6	<0.0001
MLR	0.72	1.06	0.33 ± 0.26	−0.17 to 0.84	0.19
HPR	0.04	0.05	0.008 ± 0.011	−0.01 to 0.03	0.47

**Table 3 medicina-59-00021-t003:** ROC curves results of the novel biomarkers and inflammatory standard tests.

	AUC	95% CI	Cut-off-Point	Sensibility	Specificity	*p* Value
NLR	0.77	0.73–0.80	>8.39	73.17%	70.11%	<0.0001
PLR	0.68	0.64–0.74	>201.4	63.41%	64.14%	<0.0001
CRP	0.72	0.67–0.77	>3.43	68.90%	65.29%	<0.0001
Fibrinogen	0.70	0.65–0.75	>349.2	68.29%	61.51%	<0.0001
ESR	0.66	0.61–0.72	>17.5	59.70%	68.48%	<0.0001

## Data Availability

Data supporting reported results can be found at the following email addresses: alexandru.baetu@gmail.com and cardoneanu.anca@gmail.com.

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
