# Peer review of "Predictors of Complicated Appendicitis with Evolution to Appendicular Peritonitis in Pediatric Patients"

_medicina, 2022, doi:10.3390/medicina59010021_

Round 1

Reviewer 1 Report

Authors, I really appreciate your efforts for retrospective data analysis but as you, yourself mentioned in the limitations, more observational multi-center prospective studies are required to access the utility of NLR and PLR in the form of an algorithm. This could also be the reason why NLR was not adopted in the updated 2020 WSES guidelines. NLR is widely used in clinics around the world for the diagnosis of acute abdominal infection in children. I personally did not find your retrospective study very innovative, although the data can be useful in future studies. In my opinion, multiple biomarkers, not just NLR or PLR could be useful in determining acute appendicitis in children.

Author Response

Thank you very much for your kind review. We will keep in mind your suggestions while we continue our research in this field.

Reviewer 2 Report

The statistics should be improved, as for now it is not clear if you talking about parametric or non-parametric values with the mean or median values.

The introduction and discussion parts are way too long and not relevant to the specific findings. It is not necessary to write all the information about appendicitis from radiological tests to the clinical tests if you are not investigating or make a connection to your findings.  

As the number or patients involved are large enough it would be more interesting and useful to make the connection of NLR and PLR with PAS score or CRP levels, age of the patient and so on. As it self sensitivity and specificity of the biomarkers are low. 

Author Response

The statistics should be improved, as for now it is not clear if you talking about parametric or non-parametric values with the mean or median values.

R: Thank you for your suggestion, we will work on improving it. We used parametric tests and mean values.

The introduction and discussion parts are way too long and not relevant to the specific findings. It is not necessary to write all the information about appendicitis from radiological tests to the clinical tests if you are not investigating or make a connection to your findings.

R: We will modify the manuscript accordingly to your suggestion, removing the unnecessary paragraphs.

As the number or patients involved are large enough it would be more interesting and useful to make the connection of NLR and PLR with PAS score or CRP levels, age of the patient and so on. As it self-sensitivity and specificity of the biomarkers are low. 

R: Thank you. While we continue our research in this field, we will keep in mind and apply your suggestion.

Reviewer 3 Report

Dear authors,

Overall a well constracted paper. 

Author Response

Thank you for your kind review!

Reviewer 4 Report

1. The abstract does not contain all necessary elements (e.g. results are missing).

2. The authors could freely exclude from the introduction the paragraph regarding radiological techniques (lines 87-95) for the diagnosis of acute appendicitis as they are not the subject of the research.

3. I am not sure that the aim of this study was to investigate ultrasonography features.

4. In the research methodology, you have classified the group of gangrenous and necrotic appendicitis as uncomplicated, and in the introduction you stated that WSES guidelines categorize them as complicated. This should be corrected in methodology and results, as well.

5. Results should be written in the past tense throughout.

All patients from the study had appendicitis, but one group has complicated and the other uncomplicated appendicitis. That terminology should be used.

6. Again, I would exclude the lines 168-172 from the results.

7. " Negative appendectomy rated may reach up to 20-30% [25]. 245,246 lines -this information should be checked in the literature.

8. The discussion should be shortened. The parts that are not relevant for this research should be removed. Not all the results were commented on in the discussion.

9. The conclusion is too broad. It should emphasize only certain statements.

Author Response

  1. The abstract does not contain all necessary elements (e.g. results are missing).

R: Thank you for your observation, we will modify it to include all information needed.

  1. The authors could freely exclude from the introduction the paragraph regarding radiological techniques (lines 87-95) for the diagnosis of acute appendicitis as they are not the subject of the research.

R: We will consider your suggestion and remove the paragraph from the manuscript.

  1. I am not sure that the aim of this study was to investigate ultrasonography features.

R: It is not the aim of the study, it only helps to accomplish the main aim. The purpose of this study is to identify the predictors that may aid physicians in timely identifying pediatric patients diagnosed with acute appendicitis at risk for developing complicated appendicitis with evolution to appendicular peritonitis. We will modify the manuscript to clear the information. Thank you for your observation!

  1. In the research methodology, you have classified the group of gangrenous and necrotic appendicitis as uncomplicated, and in the introduction you stated that WSES guidelines categorize them as complicated. This should be corrected in methodology and results, as well.

R: We will correct the information, thank you!

  1. Results should be written in the past tense throughout. All patients from the study had appendicitis, but one group has complicated and the other uncomplicated appendicitis. That terminology should be used.

R: We will modify it accordingly to your observation.

  1. Again, I would exclude the lines 168-172 from the results.

R: We will consider your suggestion, thank you!

  1. " Negative appendectomy rated may reach up to 20-30% [25]. 245,246 lines -this information should be checked in the literature.

R: Negative appendectomy rates varies among literature. We will check this information in more sources and modify it.

  1. The discussion should be shortened. The parts that are not relevant for this research should be removed. Not all the results were commented on in the discussion.

R: Thank you, we will review and modify the manuscript as you suggest.

  1. The conclusion is too broad. It should emphasize only certain statements.

R: We will modify it as well. Thank you for your kind suggestions.

Round 2

Reviewer 1 Report

Thank you authors for making minor changes to the manuscript. In my opinion, the two pictures showing intraoperative acute complicated appendicitis and ultrasonography could be removed from the manuscript as everyone is aware of their relevance.

Author Response

Thank you for your review! We removed the figures as you suggested.